# OpenReview forum: "mmBERT: A Modern Multilingual Encoder with Annealed Language Learning"
_ICML.cc/2026/Conference — ICML 2026 regular_

### Official Review · Reviewer_nKJK · 2026-03-08

**Soundness:** 3
**Presentation:** 3
**Significance:** 3
**Originality:** 2
**Overall Recommendation:** 5
**Confidence:** 4

**Summary:**

This paper introduces mmBERT, a modern alternative to multilingual BERT models that uses better training recipe and larger text corpus. It relies on modernBERT architecture trained on 3T tokens of multilingual text in over 1800 languages. Authors also introduce Annealed Language Learning, a novel approach for training on a multilingual dataset.

**Compliance With Llm Reviewing Policy:**

Affirmed.

**Final Justification:**

I did not have major concerns regarding this work. I believe monolingual comparisons would be valuable, but the authors point out that it is difficult to make a fair comparison between monolingual and multilingual models due to dedicated vocabulary and data collection. I believe this is a reasonable justification for not doing it. Therefore, I am increasing my score.

I would like to reiterate that this is an invaluable applied work and should not be judged based on its theoretical contributions. It deserves a spot in ICML.

**Key Questions For Authors:**

1. How do you account for differences in quality across the multilingual benchmarks used in this work?
2. How does mmBERT compare against monolingual models in low-resource languages?

**Limitations:**

Yes.

**Strengths And Weaknesses:**

Strengths:
* The paper has clearly established goals and achieves them. Experiments are well-designed and demonstrate success of mmBERT. The text, figures, and tables all are easy to follow. Authors include all relevant experimental results, including negative ones.
* Existing multilingual encoder-only models were mostly outdated. This work is well-timed and closes a significant gap. Although the paper contains limited algorithmic novelty, its significance cannot be denied.
* mmBERT outperforms mDistilBERT and Multilingual MiniLM in most multilingual benchmarks. Despite being trained on a multilingual dataset, its performance on English is comparable to that of ModernBERT, which is a monolingual model.

Weaknesses:
* Author don't compare performance of mmBERT with monolingual BERT-class models in low-resource languages. There are multiple examples of such models, and it is important to answer the question of whether multilingual models are indeed useful in low-resource languages. Although English results are available, I would suggest comparing mmBERT performance in at least several other languages, using models such aLLMA [1] and AraBERT [2]. Even more robust approach would be to train monolingual BERT models with the same architecture and training data as mmBERT.
* WikiANN benchmark, which is used in this work, contains a significant amount of labeling errors. While this may fall outside the scope of this work, I would like to note this fact for reference.
* Typo in line 373: "the the"

---

> ### Author Rebuttal · Authors · 2026-03-31
>
> Thank you for the review of our paper! We appreciate that you recognize our work by calling the experiments “well-designed” and that the system “closes a significant gap”.
>
> > *monolingual comparisons*
>
> This is an interesting question that we considered, but decided against for a few reasons. It is difficult to make a **fair** comparison between a massively multilingual system and a monolingual system. Monolingual systems can have dedicated vocabularies and are often different model sizes or depend on data availability. Solving this difficult evaluation on a case by case basis seemed brittle. For very low resource tail languages, adequate evals likely do not exist. That said, some results show that our model compares favorably against monolingual models. If you are very interested in this, we may be able to explore an anonymity preserving way to share these results during the rest of the rebuttal period.
>
> Training monolingual or less multilingual models (such as region specific encoders) is certainly a use case that we believe in. We hope our release of artifacts and recipes will enable this.
>
> > WikiANN benchmark, which is used in this work, contains a significant amount of labeling errors. While this may fall outside the scope of this work, I would like to note this fact for reference.
>
> We agree that underlying benchmark issues fall out of scope, but thank you for raising this! We used this eval, among others, as it is part of the XTREME benchmark. As other reviewers have noted, massively multilingual evaluation can be difficult. We will keep this in mind for future iterations of our multilingual work, and hope that future work in this area can develop modernized massively multilingual evaluations.
>
> > Typo.
>
> We will correct this!
>
> Q1. As our work has the highest degree of multilinguality compared to related work that we know of, massively multilingual evaluation is difficult. As you noted, there can be quality differences. One interesting future approach would be to experiment with aggregate benchmarks such as the “core” evals used in LLM work such as Olmo and DCLM. These can be weighted or calibrated (e.g. https://www.databricks.com/blog/calibrating-mosaic-evaluation-gauntlet). However, given the relative lack of massively multilingual benchmarks, we were hesitant to modify or selectively exclude evals from benchmarks. As noted in the limitations, we depend on “upstream datasets for language identification quality”. Another note regarding eval quality could also be added.
>
> Q2. Discussed above!

---

> > ### Author Rebuttal · Reviewer_nKJK · 2026-04-01
> >
> > The authors have addressed my concerns. Although they did not conduct additional experiments, they provided a reasonable justification. I am therefore increasing my score.

---

> > > ### Author Response · Authors · 2026-04-07
> > >
> > > Thanks for your feedback through the rebuttal and for raising your score in light of our rebuttal comments!

---

### Official Review · Reviewer_TnZv · 2026-03-10

**Soundness:** 3
**Presentation:** 3
**Significance:** 3
**Originality:** 2
**Overall Recommendation:** 4
**Confidence:** 4

**Summary:**

This paper introduces mmBERT, a modern multilingual encoder model. The mmBERT is based on the modern architecture of ModernBERT and the tokenizer of Gemma2. The training data is curated from open-source multilingual data, which contains 3T tokens and over 1800 languages.

To train the model, the authors propose a three-phase training schedule, including pre-training, mid-training and decay phase, with an annealed language schedule that increases the number of languages in each phase. Additionally, they also apply the techniques including inverse mask schedule and model merging.

The experiments shows that mmBERT outperforms previous multilingual encoders and is slightly worse than ModernBERT on English NLU benchmarks, but better than ModernBERT on retrieval tasks.

**Compliance With Llm Reviewing Policy:**

Affirmed.

**Final Justification:**

I am still positive for the contribution at the system level. But I dont think the contribution of training methodology is significant enough.

**Key Questions For Authors:**

No further questions

**Limitations:**

yes

**Strengths And Weaknesses:**

Strengths
1. This work builds and opens a new strong multilingual encoder mmBERT, which is a significant contribution to the community.
2. Extensive experiments demonstrate the boost performance of mmBERT compared to previous models.
3. The training data and settings are clearly elaborated for easy reproduction, and all training data is open-sourced.
4. The paper is well presented.

Weaknesses
1. Despite the effectiveness of the model, the impact of annealed language learning is not ablated, lacking theoretical or empirical results. How will it perform without annealed language learning?
2. Most of the strategies and tricks are from the existing methods.

typo: repetitive "the" in line 373.

---

> ### Author Rebuttal · Authors · 2026-03-31
>
> Thank you for the review of our paper! We appreciate you stating that our paper is well presented, making “a significant contribution to the community” with open sourced training data and “extensive experiments". We hope that others will be able to build on our open artifacts and training mixtures, especially as the community focuses on broader language data curation that could be used in future models.
>
> **W1.** Regarding ablations, we refer to the additional experiments shared in our response to reviewer BU5s
>
> **W2.** We believe that training methodology is an appropriate contribution. While we draw on related work for some of these techniques, our system demonstrates that a stronger multilingual encoder is possible. This improves on a baseline that has stood relevant for 6 years (XMLR). In particular, the phased language annealing and training schedule has no precedent and we show that it works at high scale.
>
> Typo at 373 - fixed!

---

> > ### Author Rebuttal · Reviewer_TnZv · 2026-04-03
> >
> > I am still positive for the contribution at the system level. But I dont think the contribution of training methodology is significant enough.

---

> > > ### Author Response · Authors · 2026-04-07
> > >
> > > Thanks for your recognition of our contributions and feedback on our work through this process!

---

### Official Review · Reviewer_BU5s · 2026-03-12

**Soundness:** 3
**Presentation:** 3
**Significance:** 3
**Originality:** 2
**Overall Recommendation:** 4
**Confidence:** 4

**Summary:**

This paper investigates an interesting question regarding the advancement of encoder-only multilingual models, which have been largely neglected despite their practical importance for NLU and retrieval tasks. It presents mmBERT, a modern multilingual encoder-only language model trained on 3 trillion tokens across 1833 languages. The model introduces several training strategies, including cascading annealed language learning, an inverse masking schedule, and phased language inclusion during training to improve multilingual representation learning. The paper reports better improvements over prior multilingual encoders such as XLM-R across several benchmarks including GLUE, XTREME, and MTEB. The results suggest that the proposed training strategy improves performance across both high-resource and low-resource languages.

**Compliance With Llm Reviewing Policy:**

Affirmed.

**Final Justification:**

The rebuttal has clarified some of the doubts.

**Key Questions For Authors:**

1.	The paper can provide ablation studies showing the independent contributions of: annealed language learning, inverse masking schedule, phased language inclusion? Even if expensive, ablating on a smaller scale (e.g., 300B tokens) would provide crucial evidence. What would mmBERT's performance be if trained on XLM-R's data mix but with the same setup? This would isolate the method's contribution.
2.	How sensitive is the cascading language learning approach to the temperature values (0.7, 0.5, 0.3)? Did the paper explore alternatives? What principled approach determined these values?
3.	Can the paper evaluate on additional low-resource language benchmarks (beyond Tigrinya and Faroese) to substantiate claims about low-resource language performance? Even zero-shot evaluations on diverse languages would strengthen this. Do earlier languages' performance degrade as new languages are added in subsequent phases? Can the paper provide learning curves showing per-language performance evolution through training phases?
4.	What are the approximate training compute requirements (e.g., GPU hours or FLOPs) for the full training process compared to existing similar works?
5.	The paper briefly mentions potential tokenizer limitations affecting tasks like NER and POS tagging. Why wasn't the prefix whitespace issue resolved during development? Is there a simple fix that practitioners can apply without retraining? Could the paper clarify how significant this limitation is in practice?

**Limitations:**

yes

**Strengths And Weaknesses:**

Strengths:
- The work addresses an important and timely research problem. Encoder-only multilingual models remain widely used in practical applications such as retrieval and embedding systems. Therefore, this work on encoder scaling in the multilingual setting is valuable.
- mmBERT is trained on over 1800 languages, which is significantly larger than the language coverage of earlier encoder models such as XLM-R. This scale represents a meaningful step toward broader linguistic inclusion.
- The paper presents an interesting training strategy. The proposed annealed language learning approach, where languages are gradually introduced during training and low-resource languages are added late in the training process, is a reasonable strategy for maximizing learning efficiency when data availability differs significantly across languages.
- The paper includes extensive empirical evaluation across multiple widely used benchmarks. The experiments cover classification tasks, multilingual understanding benchmarks, and retrieval tasks, and the model generally improves over prior multilingual encoders.

Weaknesses:
- While the paper introduces several training strategies, the core architecture remains largely unchanged. The model directly adopts the ModernBERT architecture. As a result, the primary contribution lies in training methodology rather than architectural innovation.
- Although the model supports a very large number of languages, the paper provides relatively little analysis of language-specific performance trends. For example, it would be valuable to understand: how performance varies across language families, how performance scales with training data availability whether the model shows improved cross-lingual transfer ability. Such analyses would help readers better understand the practical benefits of the proposed training strategy.
- The paper introduces multiple new design elements simultaneously: cascading annealed language learning, inverse masking schedule, phased language inclusion, large multilingual dataset mixture. However, the paper provides limited ablation experiments isolating the impact of each component. Without careful ablations, it is difficult to determine whether the improvements are primarily due to the proposed techniques or to the large scale of training and improved datasets.

---

> ### Author Rebuttal · Authors · 2026-03-31
>
> Thank you! We appreciate that you acknowledge our work is valuable due to the prevalence of multilingual encoders in “practical applications such as retrieval and embedding systems.”, and that our paper contains “extensive empirical evaluation”. We also appreciate that you consider it “a meaningful step toward broader linguistic inclusion.” Regarding your other specific comments:
>
> **W1**
> > “...the primary contribution lies in training methodology rather than architectural innovation.”
>
> We respectfully disagree that training methodology contributions are secondary to architectural innovation. Our training methodology is novel AND demonstrably effective in creating a better model. Constructing and open sourcing multilingual datasets is also valuable to the field.
>
> **W2: “language-specific performance trends.”**
>
> We appreciate this suggestion and agree that language specific trends are interesting. For broad transfer, we see some evidence that the languages introduced only during the decay benefit from earlier phases (e.g. Faroese may be benefiting from related languages like Norwegian and Icelandic). XNLI results in Table 5 also show which languages benefit from model scale more - English performance barely changes from small to base, while the largest changes are for Chinese, Hindi, and Arabic. We can add this to the paper and check for similar trends in other evaluations.
>
> **W3: “However, the paper provides limited ablation experiments…”**
>
> This concern was shared by a few reviewers. We note that the model has consistently better performance, giving strong evidence that the overall recipe is effective. However, we understand your desire to understand components and built the following study:
>
> ### Ablation Experiment
>
> We recognize that several reviewers have asked for further ablation experiments. Unfortunately, the scale of the dataset and language distribution make full ablations infeasible. However, we were able to construct a synthetic ablation addressing the language annealing that was key to reviewer concerns. We will include this in our final submission!
>
> In this experiment, we construct a model using English (EN), German (DE), and Turkish (TR). We simulate a low resource language by restricting TR to only 100M unique tokens (this reflects the actual EN:TR ratio). We then train two models: one with the “natural” distribution of TR throughout the training and another with the “annealed” setup, where the TR data is introduced only at the end. Total duration for both models is the same - 30B tokens. We use the last 5B tokens to decay the learning rate and (in the annealed setup) introduce the Turkish data. We use masked prediction accuracy as a simple evaluation here, measured on a held out set of equally distributed data. We observe that the annealing setup results in higher performance. See the table and [learning curve figure](https://anonymous.4open.science/r/icml-577B/masked_accuracy.jpg). The annealed setup lags until the decay since it has seen no Turkish data until the final stage of training.
>
> ```
> |          | 5B       | 10B      | 15B      | 20B      | 25B      | 30B        |
> |----------|----------|----------|----------|----------|----------|------------|
> | annealed |   51.56% |   54.77% |   56.28% |   57.24% |   57.88% | **69.32%** |
> | natural  | _58.91%_ | _62.78%_ | _64.61%_ | _65.63%_ | _66.28%_ |     68.37% |
> ```
>
> While this is a synthetic scenario, we hope that it serves as additional evidence for a core aspect of our paper. We will include a similar experiment in the full text.
>
> Q1. See response to W3. Also, we note that 300B tokens is quite expensive - around 600 H100 hours. However, we were able to use smaller compute for the ablation in W3.
>
> Q2. We were not able to ablate this approach at scale, and future work could certainly explore other strategies. We wanted an approach that shifted the distribution in favor of lower resource languages while balancing the number of repetitions. While XLMR has a very differently constructed dataset, they also use a value of 0.3 to flatten the distribution.
>
> Q3. “Can the paper evaluate on additional low-resource language benchmarks”
>
> We picked those two languages as they were both truly low resource in our data and had suitable high quality evals. Unfortunately this is a rare combination, are there particular benchmarks that you would prefer?
>
> Q4. Compute Requirements
>
> Since the model is still based on a transformer architecture, the FLOPs are the same as a transformer trained for 3T tokens. Details about practical GPU hours are present in Appendix B (roughly 40 days for the base run). XLMR was trained for around twice the duration, 6T tokens.
>
> Q5. Tokenizer prefix
>
> Unfortunately training had already begun when we realized this and the effects were unclear. We think that token based tasks like NER might be weaker than expected. https://huggingface.co/answerdotai/ModernBERT-base/discussions/48 has similar discussion

---

> > ### Author Rebuttal · Reviewer_BU5s · 2026-04-03
> >
> > Thank the authors for the rebuttal. I've increased the score after reviewing the rebuttal.

---

> > > ### Author Response · Authors · 2026-04-07
> > >
> > > Thank you for raising your score and for your feedback on our work through this process. We are glad the content and new experiments were useful for you.

---

### Official Review · Reviewer_1uBj · 2026-03-13

**Soundness:** 2
**Presentation:** 3
**Significance:** 3
**Originality:** 2
**Overall Recommendation:** 3
**Confidence:** 5

**Summary:**

This paper proposes **MMBERT**, a large-scale multilingual encoder model developed based on the **ModernBERT** architecture and optimized for multilingual learning, with a particular focus on improving performance for low-resource languages. The authors pretrain the model on approximately **3 trillion (3T) multilingual tokens** and introduce several improvements during training, including an **annealed language learning strategy**, **reverse masking scheduling**, and a **progressive increase in the number of languages** throughout the training process.

The annealed language learning strategy dynamically adjusts the sampling probabilities of different languages during training, enabling a more balanced allocation between high-resource and low-resource languages. The final model supports **1,833 languages** and can process contexts of up to **8,192 tokens**. By leveraging the efficient attention mechanism in ModernBERT, the model maintains strong performance while achieving good computational efficiency.

Experimental results show that **MMBERT** outperforms previous multilingual encoder models (e.g., **XLM-R**) on multiple multilingual benchmarks, with particularly significant improvements on low-resource language tasks. Overall, this work provides the NLP community with a large-scale multilingual encoder model as well as an improved multilingual pretraining strategy.

**Compliance With Llm Reviewing Policy:**

Affirmed.

**Final Justification:**

The rebuttal has addressed some of my concerns, and I am willing to increase my score (from 2 to 3). However, since the authors claim that the core innovation lies in the training strategy, I consider this optimization to resemble more of an engineering practice rather than a research innovation. More bluntly, this appears to be a result-driven motivation rather than a motivation-driven research solution.

**Key Questions For Authors:**

1. For the **Base version** of the model used in model merging, how do the authors select the **best checkpoint**? Is it determined based on the **validation loss during pretraining**, or based on performance on **specific languages or tasks**? Please clarify this process and provide corresponding data analysis.

2. For the **Small version** of the model in the merging experiments, the paper states that the limited parameter space of small models makes merging ineffective, but this claim lacks supporting evidence. Is the issue caused by **unreasonable performance deviations before and after merging**, or by **improper experimental configurations**? Further clarification is needed.

3. Could the authors provide a **strict controlled experiment** comparing their model with another model using the **same architecture but without the annealed language learning strategy**, in order to quantitatively measure the performance gains brought by the annealing strategy?

4. Could the authors provide a **clear and rigorous definition** for the categorization of **high-, medium-, and low-resource languages** used in the paper?

**Limitations:**

yes

**Strengths And Weaknesses:**

**Strengths:**

1. The paper proposes an **annealed language learning strategy**, which helps address the data imbalance between high-resource and low-resource languages.
2. The proposed **MMBERT** is trained on a multilingual dataset containing **3 trillion tokens** and covers more than **1,800 languages**, significantly exceeding the language coverage of many existing encoder-only models.
3. Since MMBERT is developed based on **ModernBERT**, it inherits the efficient attention implementation from ModernBERT, allowing the model to maintain strong performance while achieving high computational efficiency.
4. The paper conducts extensive experiments on multiple multilingual benchmarks, showing that **MMBERT achieves significant improvements on multilingual tasks**.

**Weaknesses:**

1. The core contribution of this paper is the **annealed language learning strategy**, which is primarily an improvement in the training strategy rather than a novel model architecture, resulting in relatively limited methodological novelty.
2. The multilingual dataset used for training is a **mixture of several datasets**, but the paper does not specify their proportions. Such mixing strategies may affect data quality and, consequently, the final model performance.
3. The paper lacks a **rigorous definition of high-, medium-, and low-resource languages**. Aside from mentioning that medium-resource languages contain more than **200M tokens**, the paper does not provide a quantitative description of the overall data distribution across the **1,833 languages**.
4. The **ablation study is insufficient**. The performance gains reported in the paper may stem from multiple factors, including the training data quality, improvements from the ModernBERT architecture, and the annealed language learning strategy. However, the paper does not provide sufficient ablation experiments to isolate the contribution of each factor.
5. Among the **1,700+ low-resource languages**, the experiments evaluate only **two low-resource languages** and then conclude that “introducing low-resource languages only during the annealing stage can significantly improve performance.” This conclusion may not be sufficiently rigorous given the limited evaluation scope.

---

> ### Author Rebuttal · Authors · 2026-03-31
>
> Thank you for your review of our paper! We appreciate that you acknowledged that our model "achieves significant improvements on multilingual tasks" and is "significantly exceeding the language coverage of many existing encoder-only models"
>
> **W1.**
>
> > The core contribution of this paper is … in the training strategy rather than a novel model architecture, resulting in relatively limited methodological novelty.
>
> We respectfully disagree - our training strategy is novel and resulted in a state of the art model. Improved training methodology should not be secondary to a new model architecture.
>
> **W2.**
> > The multilingual dataset used for training is a mixture of several datasets, but the paper does not specify their proportions. Such mixing strategies may affect data quality and, consequently, the final model performance.
>
> Are there particular details about the data mixture you find missing? We show mixture sources and sizes in Table 2, a figure of the language ratios in Figure 1, and language token breakdowns across stages in Table 10.
>
> **W3.**
> > The paper lacks a rigorous definition of high-, medium-, and low-resource languages. Aside from mentioning that medium-resource languages contain more than 200M tokens, the paper does not provide a quantitative description of the overall data distribution across the 1,833 languages.
>
> We defer to [The State and Fate of Linguistic Diversity and Inclusion in the NLP World](https://aclanthology.org/2020.acl-main.560/) (Joshi et al., ACL 2020) as a carefully reasoned argument around language resource classification. We clarify that high, medium, and low should be taken as relative to the rest of the dataset. For example, since much of the multilingual data ultimately derives from commoncrawl, low resource means that it was found in lower proportion to other written text on the internet. Regarding “the paper does not provide a quantitative description of the overall data distribution”, please see our answer above and Table 10.
>
> **W4. "The ablation study is insufficient."**
>
> We appreciate this feedback! Please see our response to reviewer BU5s for additional new ablations.
>
> **W5. evaluate only two low-resource languages**
>
> Please see our response to reviewer BU5s, Q3.
>
> Q1/Q2 “merging”.
>
> We use TIES merging for the base model and find that it did not improve performance for the small model. We evaluate the merges using benchmark tasks, not loss.
>
> Q3. Please see our response to reviewer BU5s, Ablation section.
>
> Q4. Please see our response to W3.

---

> > ### Author Rebuttal · Reviewer_1uBj · 2026-04-03
> >
> > The rebuttal has addressed some of my concerns, and I am willing to increase my score. However, since the authors claim that the core innovation lies in the training strategy, I consider this optimization to resemble more of an engineering practice rather than a research innovation. More bluntly, this appears to be a result-driven motivation rather than a motivation-driven research solution.

---

> > > ### Author Response · Authors · 2026-04-07
> > >
> > > Thanks for raising your score in light of our rebuttal, we appreciate the feedback and engagement in this process. Our work is certainly grounded in good engineering practices that let our system scale beyond other multilingual models! We hope the broader community will build on these methods and our open source artifacts.

---

### Decision · Program_Chairs · 2026-04-30

**Decision:**

Accept (regular)

**Comment:**

This paper proposes mmBERT, a massively multilingual encoder model covering over 1833 languages and trained on 3 trillion tokens. This is probably the largest ever trained encoder/LLM. This is a huge improvement over XLM-R that covers 100 languages and Glot500 covering 500 languages. They implemented some clever tricks such as annealed language learning strategy, reverse masking scheduling, and a progressive increase in the number of languages throughout the training process.

All reviewers agreed that this is a massive amount of work to train mmBERT on 1800 languages. However there are some critical issues that needs to be addressed to bring the paper to an acceptable state:
1) As reviewers 1UBj and BU5s noted, there is a disparity between the number of languages trained and evaluation conducted. Why scale to 1800 languages when you only restrict evaluation to benchmarks like XTREME that only cover 40 languages. There are several benchmarks out there with over 100 languages that are not evaluated such as Taxi1500, SIB-200, and Belebele. This is particularly disappointing, since this work does not even cite related work trying to scale to more than 500 languages such as Glot500. I think the authors should read Glot500 paper and related papers, and incorporate their style of evaluation. While it is time consuming to train an individual model for all 1500 languages covered in Taxi1500 or SIB-200, what Glot500 did was cross-lingual transfer experiments, where a model was trained on English and zero-shot evaluated on 1500 languages. This will reduce burden of evaluation.

2) I believe one major benefit of this paper is long-context feature. However, evaluation was only performed on English. Evaluating long context tasks in multilingual setting would be helpful. Details of the evaluation of mTEB v2 is not clear, is this on all languages or some?

3) Since the draft evaluate on text embedding benchmarks such as mTEBv2, fine-tuning the model further to turn it into a sentence transformer model would further strengthen the work, similar to how mGTE has a base model and a sentence embedding model.